# Does Better Diet Quality Offset the Association between Depression and Metabolic Syndrome?

**DOI:** 10.3390/nu15041060

**Published:** 2023-02-20

**Authors:** In Seon Kim, Ji-Yun Hwang

**Affiliations:** 1Department of Foodservice Management and Nutrition, Graduate School, Sangmyung University, Seoul 03016, Republic of Korea; 2Major of Foodservice Management and Nutrition, Sangmyung University, Seoul 03016, Republic of Korea

**Keywords:** depression, diet quality, metabolic syndrome, PHQ-9, healthy eating index

## Abstract

Several studies have shown that depression increases the risk of metabolic syndrome (MetS), which is often exacerbated by the fact that both exist concurrently. People with depression are more likely to have unhealthy eating habits, which can eventually trigger the development of MetS. This study was to investigate whether diet quality modifies the association between depression and MetS in a total of 13,539 Korean adults aged 19 to 80 from 2014, 2016 and 2018 Korean National Health and Nutrition Examination Surveys. Depression was assessed by the Patient Health Questionnaire-9 (PHQ-9) and subjects were divided into subgroups according to the PHQ-9 scores: normal (<5), mild (5–9), and moderate-to-severe (≥10) groups. Diet quality was measured by the Korean Healthy Eating Index (KHEI). A complex sample multiple logistic regression stratified by tertiles of KHEI scores was used to explore whether diet quality modifies an association between depression severity and metabolic syndrome. Depression severity was positively associated with the risk of MetS (*p* trend = 0.006) after adjustment for potential confounders. Only the lowest diet quality, moderately-to-severely depressed group, showed a higher risk of MetS (OR: 1.72, 95% CI: 1.24–2.40) compared to the normal group. Our results suggest that healthy diet quality could offset the positive relationship between depression and MetS in the general Korean adult population. Encouraging a healthy diet regime can improve not only physical health but also the mental state of the general public.

## 1. Introduction

Depression is a common mental disorder, and it is estimated that about 260 million adults suffer from depression [1]. A systematic analysis of the global burden of disease study showed that depressive disorder was one of the top ten causes of disability-adjusted life-years for young age groups (10–49 years of age) in 2019 [2]. Metabolic syndrome (MetS), another big health threat, is a cluster of conditions, such as increased waist circumference, dyslipidemia (reduced HDL and elevated triglyceride levels), increased blood pressure, and increased fasting blood sugar levels [3], and significant risk for cardiovascular and type 2 diabetes [4].

Although the association between depression and MetS was inconsistent depending on the degree of mood disorder in early studies [5], the latest meta-analysis showed that people with depression had a 1.48 times higher risk of MetS than those without depression [6]. Although the mechanism underlying the association between depression and MetS has not been completely understood, it could be explained by two hypotheses. First, conditions of depressive symptoms and Mets may have high pathophysiological overlaps, such as a common pathway in the stress system [7]. It involves an abnormal activation of the Hypothalamic-Pituitary-Adrenal Axis and an imbalance of the autonomic nervous system, which could contribute to insulin resistance, abdominal obesity, and dyslipidemia [8,9]. Moreover, both conditions share low-grade systemic inflammation and oxidative stress [10]. Second, the co-occurrence of depression and MetS may be mediated by unhealthy lifestyle factors, including poor diet and physical inactivity, such as a sedentary lifestyle [11]. Poor diet, in particular, has been related to both conditions in many studies [11,12,13,14,15,16,17,18,19]. Depressed people are more likely to have unhealthy dietary patterns through altered motivation, and it could, in turn, trigger the development of obesity [12]. Whereas the Mediterranean dietary pattern, one of the healthy diet regimes [13], has a protective effect against depression and has been known to decrease the risk of obesity, type 2 diabetes, and cancer [14,15,16]. 

Although diet quality may be a potential effect modifier to the association between depression and MetS, to the best of our knowledge, limited research has been done in the general population with potential confounding factors. Therefore, this study was to investigate whether diet quality modifies the association between depressive symptoms and MetS in the general Korean population.

## 2. Materials and Methods

### 2.1. Design and Data Collection

We used data from the Korean National Health and Nutrition Examination Survey (KNHANES) conducted by the Korea Disease Control and Prevention Agency (KDCA). The KNHANES employs a multi-stage stratified cluster sampling for the selection of household units among non-institutionalized residents in the Republic of Korea. The survey is comprised of a health interview, a health examination, and a nutrition assessment. Details about the survey are available on the KNHANES website [20] (https://knhanes.kdca.go.kr/knhanes/main.do, accessed on 1 March 2021).

This study included KNHANES 2014, 2016 and 2018 data that conducted the Patient Health Questionnaire-9 (PHQ-9) survey to determine depression severity. Among the participants (*n* = 23,692) aged 19 to 80, we excluded pregnant or lactating women and those who had missing information on weight, height, residential area, education level, family income, drinking, smoking, metabolic syndrome markers (or doctor diagnosis of disease), PHQ-9 scores, Korean Healthy Eating Index (KHEI), energy intake, physical activity, and history of the disease. A total of 13,539 participants were finally eligible for the analysis. 

Written informed consent was obtained from all participants. The institutional review board (IRB) of the KDCA approved the survey, and the additional IRB process for this study was also approved. (2013-12EXP-03-5C, 2018-01-03-P-Aex21, SMUIRB ex-2021-003).

### 2.2. Methods and Variables

#### 2.2.1. Socio-Demographic Factors

In this study, the demographic and socioeconomic variables of the participants were obtained, including age, sex, height, weight, residential area, household income, education level, smoking status, alcohol drinking, physical activity, and disease history. The residential area was classified into urban or rural. Education level was categorized as ≤elementary-school graduate, middle-school graduate, high-school graduate, and ≥college graduate. Household income level was categorized into four quartile groups. Smoking status was classified into two groups (non-smoker or current smoker), and alcohol drinking consumption was categorized into five (never, ≤1 drink/month, 2–4 drinks/month, 2–3 drinks/week, and ≤4 drinks/week). Physical activity was defined as ≥150 min/week in moderate exercise activities, or ≥75 min/week in vigorous exercise activities or mixing two exercise intensities (doing 1 min in vigorous exercise equals 2 min in moderate exercise). Disease history was defined as whether a subject was diagnosed with hypertension, hyperlipidemia, stroke, myocardial infarction (or heart attack), arthritis, asthma, tuberculosis, kidney failure, diabetes, cancer, heart failure, or cirrhosis of the liver.

#### 2.2.2. Assessment of Depressive Symptoms

The severity of depressive symptoms was evaluated with the PHQ-9, a depression measurement tool designed for criteria-based diagnosis of depression and mental disorders [21]. The Korean version of PHQ-9 has been tested for reliability, validity, and clinical usefulness [22,23,24] and is investigated once every two years in the KNHANES. It consists of nine questions, and the scales of questions asking the frequency of each symptom over the past two weeks are scored 0 (‘not at all’), 1 (‘several days’), 2 (‘over a week’), and 3 (‘nearly every day’). The total scores ranged from 0 to 27 and were used to classify participants into three groups according to the validation study of the Korean version of PHQ-9 [25]: normal group (<5), mild group (5–9), and moderate-to-severe group (≥10).

#### 2.2.3. Assessment of MetS

Based on the 2005 revised National Cholesterol Education Program Adult Treatment panel-III criteria [26], MetS was defined as having three or more of the followings: (1) abdominal obesity (waist circumstance (WC) ≥ 90 cm for men, ≥ 80 cm for women, according to Asian and Pacific regions); (2) elevated serum triglyceride (TG) level (TG ≥ 150 mg/dL or current drug treatment for high TG); (3) reduced HDL-cholesterol (HDL-C) (HDL-C < 40 mg/dL for men, 50 mg/dL for women); (4) elevated blood pressure (average systolic blood pressure (SBP) > 130 mmHg or diastolic blood pressure (DBP) >85 mmHg or current drug treatment for hypertension); and (5) elevated fasting blood glucose (FBG) (FBS ≥ 100 mg/dL or current drug treatment for a hypoglycemic agent or insulin).

#### 2.2.4. Assessment of Diet Intake and Diet Quality

Dietary intake was assessed using a 24-h recall method during the KNHANES to obtain individual food and nutrient consumption. Diet quality was assessed using KHEI, a validated standardized evaluation tool for overall diet quality for Korean adults, by scoring adherence to dietary guidelines [27,28,29]. It consists of 14 components with 3 categories of adequacy (8 components), moderation (3 components), and balance (3 components) [29]. The adequacy category measures whether the recommended intakes of food and nutrients based on dietary guidelines are sufficiently met: frequency of breakfast intake; mixed grain intake; total fruit intake (all fresh fruits, canned fruits, and dried fruits (except fruit juice)); fresh fruit intake; total vegetable intake (all vegetables, mushrooms, and seaweeds); vegetable intake that excludes salted vegetables such as kimchi and pickles according to the dietary guideline of ‘consume 2 to 3 sorts of vegetables except kimchi in every meal’; meat, fish, egg and bean intake; and milk and dairy product intake [29]. The moderation category includes saturated fatty acid energy intake ratio, sodium intake, and sugar and beverage energy intake ratio [29]. The balance category evaluates the balance of energy and macronutrient intake, i.e., the percentage of energy from carbohydrates and fat and energy intake [29]. More detailed information regarding the KHEI was described in the previous study [29]. Among eight adequacy components, two components, ‘having breakfast’ and ‘meat, fish, eggs, and beans intake’ are given 0–10 scores, and five other components are given 0–5 scores (maximum score: 55). In moderation components, all components are given 0–10 points (maximum score: 30), and all of three balance components are given 0–5 scores (maximum score: 15), resulting in a total score of 100 [29]. The KHEI score is outlined in the KNHANES raw data based on the 2015 Dietary Reference Intakes for Koreans [30]. The higher score calculated through each component represents participant has better diet quality. In this study, participants are divided into tertiles of diet quality based on total scores of KHEI: lowest diet quality (T1) to most optimal diet quality (T3).

### 2.3. Statistical Analysis

Due to the complex survey design of KNHANES, all statistical analyses applied integrated weights calculated to consider sampling units, stratification, and sample weights. The comparison of continuous variables was presented as means with standard error (SE) using a complex sample multiple regression analysis (Bonferroni test of multiple comparisons). The comparison of categorical variables was presented as numbers with weighted % using a chi-squared test. Potential interactions between confounders have been considered, and household income and smoking status were finally excluded due to the possibility of interaction with other confounding factors. Physical activity was included as a common confounding factor, as reported previously [31]. Multivariate analyses were adjusted for age, sex, BMI, education level, alcohol drinking, energy intake, and physical activity. Adjusted odds ratio (OR), 95% confidence interval (CI), and *p* trend of MetS according to depression groups were evaluated by a multivariable-adjusted logistic regression analysis after adjusting for confounders. To examine whether diet quality modifies the association between depressive symptoms and MetS, data were further stratified according to tertiles of KHEI scores. Furthermore, in order to identify a potential mediation effect of diet quality on the association between depressive symptoms and risk of MetS, we performed a series of hierarchical logistic regressions after adjusting for confounders. Three conditions must be met to demonstrate the mediation [32]: (1) the independent variable (X, depression severity) is significantly related to the dependent variable (Y, MetS); (2) the independent variable is significantly related to the mediator variable (M, diet quality); and (3) the mediator variable is significantly associated with the dependent variable when the independent variable (depression severity) is controlled for. If the conditions are all met, we used the formula [32] to calculate the proportion of mediation. (OR^DE^ means the direct effect of X on Y controlling for M, and OR^IE^ means the indirect effect of X on Y through M.)
$$\frac{OR^{DE}\times \left(OR^{IE}-1\right)}{\left(OR^{DE}\times OR^{IE})-1\right)}$$

Statistical analyses were performed using SPSS version 23.0 (IBM Corp., Armonk, NY, USA). Statistical significance was determined at *p*-value < 0.05.

## 3. Results 

### 3.1. Socio-Demographic Factors According to Depression Severity

We examined socio-demographic factors according to depression severity (Table 1). A total of 13,539 participants were classified as normal (*n* = 10,888, 80.4%), mildly depressed (*n* = 1905, 14.1%), and moderately-to-severely depressed (*n* = 746, 5.5%) according to PHQ-9 scores. The mean age was greater in the normal and moderate-to-severe groups than that in the mild group (*p* < 0.001). The normal group had greater proportions of males, higher education levels, household income, and non-smokers, and a lower frequency of alcohol drinking than the other two groups (all; *p* < 0.001). The normal group had a higher BMI than the mild group (*p* = 0.002) and had the highest energy intake (*p* < 0.001), and had the lowest proportion of disease history (*p* < 0.001) than the other two groups. 

### 3.2. Diet Quality Measured by KHEI According to Depression Severity Groups

Total scores of KHEI and breakfast item scores were the highest in the normal group and lowest in the moderately-to-severely depressed group (both *p* < 0.001, Table 2). The normal group had greater scores on intakes of mixed grains (*p* = 0.021), vegetable intakes excluding kimchi and pickled vegetables (*p* = 0.001), and energy from sweets and beverages (*p* < 0.001) and fat (*p* = 0.048) than those of depressive groups. The moderately-to-severely depressed group had the lowest scores of consumption of total fruit (*p* < 0.001), fresh fruit (*p* = 0.002), total vegetable (*p* < 0.001), and meats/fishes/eggs/beans (*p* < 0.001) and total energy (*p* < 0.001). Only scores of sodium intake were greater in the depressed groups than in the normal group (*p* = 0.038). Total scores of KHEI and most of each item score of adequacy, moderation, and balance were decreased as depressive symptoms increased (*p* trend values raged from < 0.001 to 0.045). Only scores of sodium intake increased as depression severity increased (*p* = 0.015).

The moderately-to-severely depressed group had lower intakes of polyunsaturated fatty acid (*p* = 0.035), *n*-6 fatty acid (*p* = 0.018), and potassium (*p* = 0.001) than the other two groups (Table 3). The mean intakes of total sodium (*p* = 0.026, *p* trend = 0.012) and riboflavin (*p* = 0.035) were lower in the moderately-to-severely depressed group than in the normal group. Intakes of protein, dietary fiber, phosphate, sodium, and potassium were decreased as depression severity increased (*p* trend ranged from < 0.001 to 0.040).

### 3.3. KHEI Scores and Nutrient Intakes According to Tertiles of KHEI scores

Scores of all individual components of KHEI were significantly increased from T1 to T3 (all; *p* < 0.001) except for sodium intake (Table 4). T2 had the lowest score versus the other two groups (*p* < 0.001). As described [29], the difference in scores needs to be interpreted as a KHEI scoring system according to a dietary guideline for Koreans. For example, a 1.7-point difference between T3 and T1 in mixed grain intake means that T3 consumes 0.1 serving/day more mixed grains than T1 (maximum score: ≥ 0.3 serving/day). 

Intakes of all nutrients except for polyunsaturated fatty acid and retinol increased as diet quality increased (Table 5). T1 had the lowest intakes of most nutrients than the other two groups but higher intakes of sodium than T3 (*p* < 0.001).

### 3.4. MetS Parameters According to Depression Severity Groups

The mean of TG was the lowest (*p* < 0.001), but that of DBP (*p* = 0.008) was the highest in the normal group than the other two depressive groups (Table 6). The normal group had lower HDL-C compared to the mildly depressed group (*p* = 0.015). As the PHQ-9 score increased, levels of TG (*p* trend < 0.001), HDL-C (*p* trend = 0.020), and FBG (*p* trend = 0.026) increased, but DBP (*p* trend = 0.003) decreased. There were no significant differences in WC and SBP across the depression severity. 

### 3.5. Adjusted OR (95% CI) for MeSe According to Depression Severity Groups

The moderately-to-severely depressed group had a 47% higher risk of MetS, a 30% higher risk of hypertriglyceridemia, and a 23% higher risk of hyperglycemia compared to the normal group (Table 7). The mildly depressed group had a 14% higher risk of hypertriglyceridemia and a 16% lower risk of low HDL-C than the normal group. As PHQ-9 scores increased, risks of MetS (*p* trend = 0.006), hypertriglyceridemia (*p* trend = 0.003), and low HDL-C (*p* trend = 0.001) increased. However, an inverse relationship was observed between depression and high blood pressure (OR = 0.84, 95% CI: 0.73–0.97, *p* trend = 0.034). 

### 3.6. Effect Modification of Diet Quality on Association between Depression and MetS

The positive associations between depression severity and risk of MetS were only observed in the lowest tertile group of diet quality (Table 8). In the lowest tertile (T1), the OR for MetS was significantly higher in the moderate-to-severe group compared to the normal group (OR = 1.72, 95% CI: 1.24–2.40, *p* trend = 0.009). Consistently with MetS, subjects with moderate-to-severe depressive symptoms had an increased risk of abdominal obesity (OR = 1.70, 95% CI: 1.04–2.81, *p* trend = 0.040). Moreover, a clear linear trend was found in the risk of low HDL-C (OR = 1.35, 95% CI: 1.06–1.71 in the mild group; OR = 1.51, 95% CI: 1.11–2.03 in the moderate-to-severe group; *p* trend < 0.001). In the middle tertile (T2), a similar result was found in the risk of low HDL-C in the mild group (OR = 1.24, 95% CI: 1.01–1.52), but an inverse relationship was observed in the risk of high blood pressure (OR = 0.57, 95% CI: 0.41–0.80 in moderate-to-severe group; *p* trend = 0.003). In the highest tertile (T3) of diet quality, the positive relationship between depressive severity and risk of hypertriglyceridemia was only observed (*p* trend = 0.010). These clear effect modifications of diet quality on positive associations between depression and MetS were summarized in Figure 1. A healthy diet may offset the associations between depressive severity and risk of MetS based on our results.

Whether diet quality mediated the association between depression severity and risk of MetS was also tested, controlling for the potential confounders (Table 9, Figure 2). However, no mediation effect existed, including sub-categories of KHEI (Appendix A). Furthermore, a potential mediation effect of fulfilling each component of KHEI was examined (Appendix A), and mediating effects existed for vegetable subcomponents (proportion of mediation: total vegetable intake 63.3%; vegetable intake excluding kimchi and pickled vegetables 66.3%). Therefore, scores of KHEI and nutrient intakes were compared according to fulfilling both vegetable intake subcomponents (Appendix A). Interestingly, those having full scores of vegetable intakes had lower scores of intakes of sodium and milk and dairy product and more sodium intakes than their counterpart. The positive associations between depression severity and risk of MetS were only observed in those without fulfilling vegetable intake subcomponents (Appendix A). However, an inverse association between depression severity and risk of high blood pressure existed in those with full scores of vegetable intake subcomponents (*p* trend = 0.047).

## 4. Discussion

We found a strong effect of modification of diet quality on the positive relationship between depression and MetS using KNHANES data (2014, 2016, and 2018). Only in the lowest diet quality group did moderately-to-severely depressed individuals have a higher risk of MetS compared to normal ones. Moreover, subjects with moderate-to-severe depressive symptoms had increased risks of abdominal obesity, and both depressed groups had an increased risk of low HDL-C compared to the normal group. Our results indicate that a healthy diet may offset the associations between depressive severity and the risk of MetS.

Most of the scores of KHEI subcomponents were lower in the two depression groups than in the normal group, but it is worth noting that the score of sodium intake was greater due to a lower intake of sodium in the two groups. Furthermore, an inverse association between depression severity and risk of high blood pressure existed in the middle tertile group of diet quality and in those with full scores of vegetable intake subcomponents. The same inverse association has been found in the cross-sectional studies in the general US female population using National Health and Nutrition Examination Survey [33] and elderly Japanese women [34]. Excessive sodium intake increases the risks of hypertension and cardiovascular events due to negative physiological effects on the body system [35]. Due to the nature of cross-sectional studies including ours, it is not clear whether the reduction in sodium intake increases stress or stress increases sodium intake. However, considering mediating effects of fulfilling vegetable subcomponents of KHEI, stress may increase sodium intake via high vegetable intake that, eventually, increases the risk of MetS. Further understanding between depression, diet quality, and MetS is needed. 

Consistently with previous meta-analysis studies [6,36,37,38], our findings showed that depressive people had a higher risk of MetS than non-depressive ones. For the components of MetS, significant associations did not exist between depression and abdominal obesity, as shown in one previous study using KNHANES 2007–2013 among women [39]. However, a randomization study with European ancestry identified that genetically predicted depression is a risk factor for MetS, abdominal obesity, hypertension, and hypertriglyceridemia but not a risk factor for low HDL-C and hyperglycemia [40]. These incompatible results may be due to population traits or residual confounding depending on confounding factors used in each study. Therefore, further studies need to investigate the association between depression and MetS with a large-scale prospective cohort study design or meta-analysis after controlling for potential confounders to minimize residual confoundings.

Among the components of MetS, no association between depressive symptoms and hyperglycemia existed. Although hyperglycemia plays a crucial role in MetS development as an indicator of insulin resistance, it is known that elevated TG levels are also correlated with insulin resistance in individuals with normal glucose levels [41]. Insulin resistance may cause nonalcoholic fatty liver disease by the elevated accumulation of free fatty acids due to hypertriglyceridemia in the liver [42,43] before type 2 diabetes develops [44,45]. Therefore, individuals with dyslipidemia need to be treated carefully, even though their blood glucose levels are still normal.

We found an inverse association between the risk of depression and high blood pressure, although subjects with current medication for hypertension classified as high blood pressure may have an almost normal range of blood pressure. Previous studies investigating the association between depressive symptoms and blood pressure have shown conflicting results [46,47]. A cross-sectional study in Iran reported that depression was associated with only high blood pressure among MetS components in elderly men [46]. On the other hand, an inverse association between blood pressure and depression was also reported in that subjects with depression and anxiety showed lower SBP and less hypertension than normal subjects [47]. Several mechanisms have been suggested for the inverse relationship. First, the depressed may use antihypertensive drugs to lower blood pressure levels, although we excluded subjects with cardiovascular diseases, including heart failure, heart attack, and hypertension. Second, conversely, low blood pressure induces depressive symptoms via somatic symptoms and fatigue [48]. Third, neuropeptide Y might be an important common underlying factor that independently explains low blood pressure and depressive symptoms [49,50]. 

Our results suggest that diet quality may play a pivotal role in modifying the association between depressive symptoms and MetS. A previous study has also reported that diet and physical activity explained 23 percent of the association between the two conditions using a path analysis after controlling for age, sex, education and income [51], although methodological and population differences may exist. Among several inflammatory and metabolic processes, increased production of proinflammatory cytokines and endothelial dysfunction is the most important potential link between depression and cardiometabolic disorders [52,53]. Cytokines can contribute directly to MetS, causing insulin resistance [54] and promoting depression-like behavior during chronic stress [55]. The animal model study [46] suggests that high-fat-fed rats accompany anxiety and depressed behavior with an increase in glucose intolerance and inflammatory cytokines such as IL-1, IL-6, and TNFlevels; they also found that treating with a P2 × 7 receptor antagonist, which blocks activation of the inflammasome, can block the behavioral abnormalities induced by a high-fat diet [56]. This result strengthens the common link between metabolic, psychiatric disorders and nutrition. In addition, inhibition of the expression of a brain-derived neurotrophic factor due to low-grade inflammatory conditions and endothelial dysfunction can mediate progressive neurological dysfunction [57]. A healthy diet with adequate balance and moderation might prevent and improve inflammation condition as well as endothelial dysfunction.

A potential mediation effect of diet quality on the association between depressive symptoms and risk of MetS was also examined. No mediating effect was found in the overall diet quality measured by a total score of KHEI, but substantial mediating effects were found according to whether a vegetable intake guideline was fulfilled. Although several studies [58,59], including meta-analyses [60,61], have reported an inverse association between vegetable intakes and the risk of MetS, a recent study in Japan has reported no association [62]. As mentioned previously, the greater the vegetable intake, the greater the sodium intake in our subjects (3436.02 mg vs. 4413.68 mg, as shown in Appendix A). A high sodium intake has been known as a predictor of the development of MetS [35], especially hypertension [63]. In addition, subjects with full scores of vegetable intake had lower scores of intakes of milk and dairy product and total energy. Taken together, even though the mediation effects of fulfilling vegetable recommendations existed, the recommendation for improving overall diet quality is more favorable for patients with depression or MetS before incontrovertible evidence is proved. 

Our findings could be interpreted that the low quality of diet not only significantly increases the risk of MetS in depressive individuals, but also a healthy diet alleviates the risk of MetS. It has been suggested that the diet intervention for MetS might be useful for getting better depressive symptoms [64] since MetS and depressive symptoms seem to share common physiopathological mechanisms [7,64]. A previous study has reported that a dietary intervention for weight loss was associated with decreased depressive symptoms in subjects with MetS [65]. Moreover, several studies have proposed that the Mediterranean diet, one of the healthy diet standards [13], protects people against depressive symptoms as well as MetS and its related chronic diseases [14,15,16]. Furthermore, a strong relationship between a proinflammatory diet and depression has also been found in subjects with cardiometabolic diseases [66].

Our study has several limitations. We could not make causality of the association because the KNHANES data used in this study is a cross-sectional survey. Future studies should utilize available longitudinal data to investigate mediating effects of fulfilling vegetable intakes on the association between depressive severity and risk of MetS and an observed inverse association between depression severity and risk of high blood pressure. The proportion of moderately-to-severely depressed subjects is 5.5%, showing a large difference in proportions of normal and moderately depressed subjects, although the prevalence is similar to that of lifetime major depressive disorders of 7.7% assessed in the general Korean population from the Korea National Mental Health Survey in 2021 [67]. If data from a prospective cohort or a prospective randomized controlled study are available in the future, a potential probability of confounding could be eliminated via randomly extracting normal and depressive groups. The degree of depression assessed by the self-reported PHQ-9 may be likely to be biased and prone to errors, although it was validated for reliability and clinical usefulness as a screening tool [22,23,24]. Lastly, the lack of data about medication use may also be a major limitation. Metformin, a major medication for type 2 diabetes, has been proposed to treat comorbid depression and diabetic patients [68]. Several studies have suggested that antidepressants can induce (Mirtazapine (SNRI), TCAs amitriptyline and nortriptyline) and reduce (selective Serotonin re-uptake inhibitors) weight gain as a side effect [69,70,71], although results vary depending on the duration of exposure and types of medication [72]. Therefore, information on medication needs to be included in future studies. Despite these limitations, to the best of our knowledge, our study is the first study that examined the effect of modification of diet quality on the association between depression and MetS using standardized indicators with potential confounders. Moreover, we cover the general adult population with nationally representative samples. Findings from our study could be used as meaningful basic data to explore effect modification factors on the association between depression and MetS for a general population in the future. 

## 5. Conclusions

Our results suggest that healthy diet quality could offset the positive relationship between depression and MetS in the general Korean adult population. Encouraging a healthy diet regime can improve not only physical health but also the mental state of the general public, and our findings may provide an evidence-based message for people with depression to eat healthily. 

## Figures and Tables

**Figure 1 nutrients-15-01060-f001:**
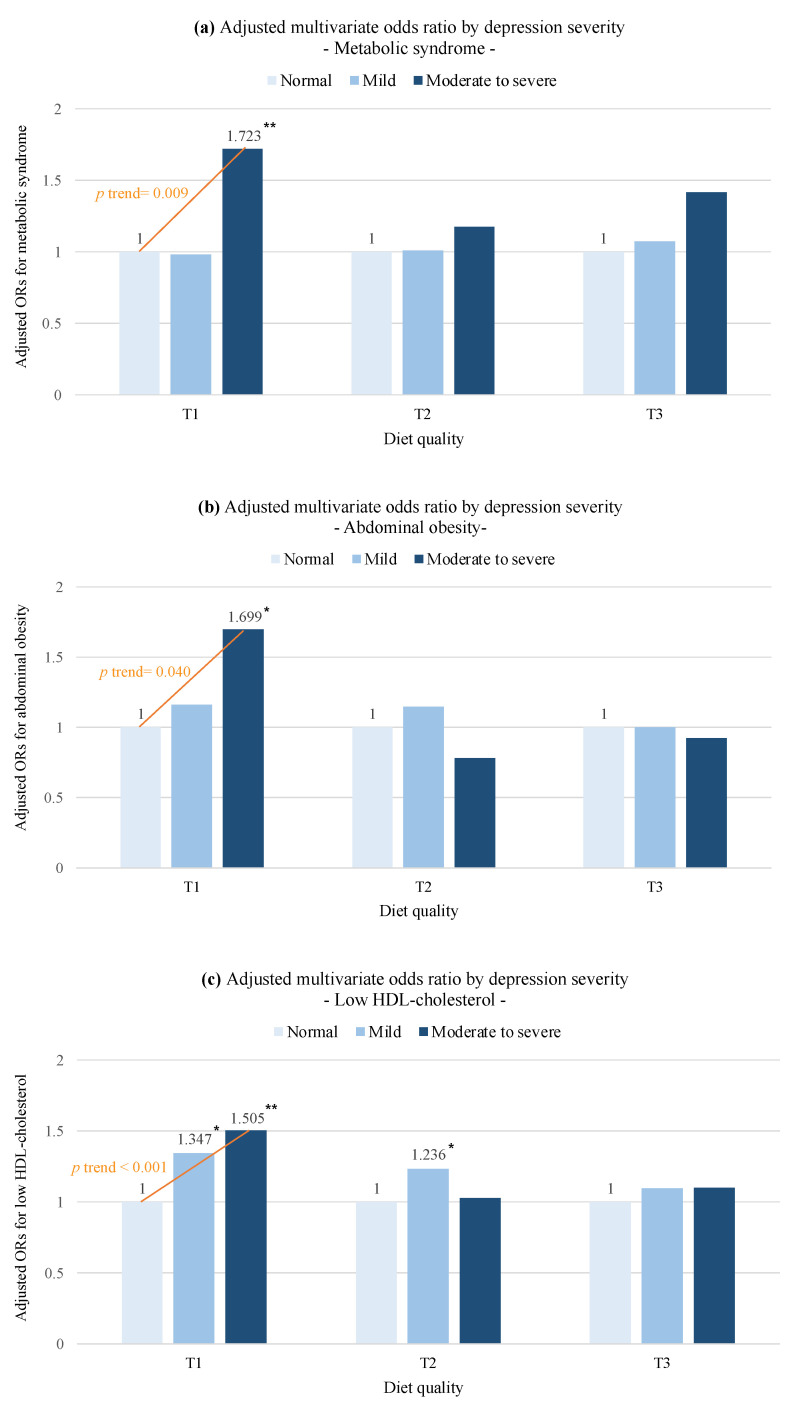
Adjusted odds ratios of metabolic syndrome (**a**), abdominal obesity **(b**), and low HDL-cholesterol (**c**) according to depression severity after stratified by diet quality based on results of Table 5. The reference group was a normal group of each diet quality level. PHQ-9: Patient Health Questionnaire-9. PHQ-9 depression severity was divided by total scores of PHQ-9: normal, mild, moderate, to severe. KHEI: Korean Healthy Eating Index. Diet quality level was divided into tertiles by total scores of KHEI: Low (T1), Medium (T2), and High (T3). * *p* < 0.05, ** *p* < 0.01.

**Figure 2 nutrients-15-01060-f002:**
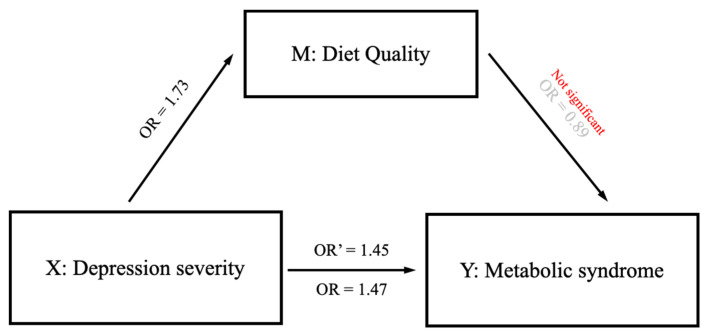
A framework of mediation analysis. OR: odds ratios. X: independent variable. M: mediating variable.

**Table 1 nutrients-15-01060-t001:** Socio-Demographic factors of subjects according to depression severity.

		PHQ-9 Depression Severity			
Variables		Normal (*n* = 10,888)	Mild (*n* = 1905)	Moderate to Severe (*n* = 746)	*p*-Value
Age, years		47.09 ± 0.26 ^a^	44.36 ± 0.47 ^b^	47.34 ± 0.79 ^a^	<0.001
Male		4879 (51.57)	596 (37.10)	204 (31.54)	<0.001
Urban residents		8892 (85.37)	1540 (83.95)	583 (83.14)	0.225
Education level	≤Elementary school	2103 (13.62)	432 (16.40)	266 (26.54)	<0.001
	Middle school	1137 (8.73)	199 (9.04)	84 (9.31)	
	High school	3603 (36.87)	634 (36.88)	209 (33.04)	
	≥College graduate	4045 (40.78)	640 (37.68)	187 (31.12)	
Household Income	Low	1769 (13.16)	420 (18.00)	287 (33.63)	<0.001
	Low-middle	2629 (23.40)	496 (27.24)	195 (24.84)	
	Middle-high	3142 (30.76)	502 (27.94)	158 (22.23)	
	High	3348 (32.69)	487 (26.82)	106 (19.30)	
Current smokers		1787 (19.74)	364 (23.10)	177 (27.71)	<0.001
Alcohol drinking	Never	2936 (22.68)	521 (22.91)	271 (30.49)	<0.001
	<1 time/month	3140 (28.81)	554 (29.46)	190 (27.60)	
	2–4 times/month	2447 (25.00)	424 (25.06)	117 (17.86)	
	2–3 times/week	1653 (16.87)	275 (15.49)	91 (13.36)	
	≥4 times/week	712 (6.65)	131 (7.07)	77 (10.69)	
Body mass index, kg/m		23.95 ± 0.04 ^a^	23.53 ± 0.11 ^b^	23.91 ± 0.19 ^ab^	0.002
Physical inactivity		5123 (50.62)	869 (49.59)	310 (46.57)	0.190
Energy intake, kcal		2087.06 ± 12.48 ^a^	1990.87 ± 25.92 ^b^	1869.10 ± 48.84 ^c^	<0.001
No disease history		6516 (54.79)	1283 (60.96)	574 (73.43)	<0.001

PHQ-9: Patient Health Questionnaire-9. PHQ depression severity was divided by PHQ total scores (normal: < 5, mild: 5–9, moderate-to-severe: ≥10). Values are presented as mean ± standard error or frequency (weighted %). One *p*-value was obtained from a complex sample multiple regression analysis for continuous variables; two *p*-values were obtained from a chi-squared test for categorical variables^; a–c^ were Significantly different at *p* < 0.05, ascertained by Bonferroni’s test.

**Table 2 nutrients-15-01060-t002:** Diet quality was measured by the Korean healthy eating index of subjects according to depression severity.

	PHQ-9 Depression Severity				
	Normal (*n* = 10,888)	Mild (*n* = 1905)	Moderate to Severe (*n* = 746)	*p*-Value	*p* Trend
Total score	63.46 ± 0.19 ^a^	61.11 ± 0.35 ^b^	58.46 ± 0.62 ^c^	<0.001	<0.001
Adequacy item					
Having breakfast	7.27 ± 0.06 ^a^	6.42 ± 0.11 ^b^	6.20 ± 0.19 ^c^	<0.001	<0.001
Mixed grain intake	2.08 ± 0.03 ^a^	1.86 ± 0.06 ^b^	1.86 ± 0.10 ^b^	0.021	0.007
Total fruit intake	2.22 ± 0.03 ^a^	2.14 ± 0.06 ^a^	1.87 ± 0.10 ^b^	<0.001	<0.001
Fresh fruit intake	2.41 ± 0.03 ^a^	2.32 ± 0.06 ^a^	2.05 ± 0.10 ^b^	0.002	0.001
Total vegetable intake	3.55 ± 0.02 ^a^	3.33 ± 0.04 ^a^	3.10 ± 0.07 ^b^	<0.001	<0.001
Vegetable intake excluding kimchi and pickled vegetables	3.28 ± 0.02 ^a^	3.06 ± 0.04 ^b^	2.90 ± 0.07 ^b^	0.001	<0.001
Meats/fishes/eggs/beans intake	7.25 ± 0.04 ^a^	7.05 ± 0.09 ^a^	6.16 ± 0.17 ^b^	<0.001	<0.001
Milk/dairy product intake	3.36 ± 0.06	3.26 ± 0.12	2.91 ± 0.19	0.132	0.045
Moderation item					
Energy from saturated fatty acid	7.45 ± 0.04	7.34 ± 0.11	7.48 ± 0.18	0.913	0.934
Sodium intake	6.39 ± 0.05 ^a^	6.73 ± 0.09 ^ab^	7.26 ± 0.16 ^b^	0.038	0.015
Energy from sweets/beverages	9.18 ± 0.03 ^a^	8.90 ± 0.07 ^b^	8.80 ± 0.13 ^b^	<0.001	<0.001
Balance item					
Energy from carbohydrate	2.53 ± 0.03	2.48 ± 0.06	2.22 ± 0.09	0.152	0.054
Energy from fat	3.38 ± 0.02 ^a^	3.23 ± 0.06 ^b^	3.07 ± 0.09 ^b^	0.048	0.019
Total energy intake	3.11 ± 0.03 ^a^	2.99 ± 0.06 ^a^	2.58 ± 0.10 ^b^	<0.001	<0.001

PHQ-9: Patient Health Questionnaire-9. PHQ depression severity was divided by PHQ total scores (normal: <5, mild: 5–9, moderate-to-severe: ≥10). Values are presented as mean ± standard error. ^a–c^ Significantly different at *p* < 0.05 by Bonferroni’s test. Adjusted for age, sex, body mass index, education level, alcohol drinking consumption, energy intake, disease history, and physical activity.

**Table 3 nutrients-15-01060-t003:** Nutrient intakes of subjects according to depression severity.

	PHQ-9 Depression Severity				
	Normal (N = 10,888)	Mild (*n* = 1905)	Moderate to Severe (*n* = 746)	*p*-Value	*p* Trend
Protein, g	75.20 ± 0.53	71.39 ± 1.11	64.99 ± 2.28	0.083	0.040
Fat, g	48.05 ± 0.48	47.17 ± 1.07	41.13 ± 1.55	0.147	0.967
Saturated fatty acid, g	14.98 ± 0.17	14.55 ± 0.36	12.85 ± 0.51	0.174	0.508
Monounsaturated fatty acid, g	15.47 ± 0.17	14.99 ± 0.39	12.92 ± 0.57	0.088	0.506
Polyunsaturated fatty acid, g	12.26 ± 0.13 ^a^	12.33 ± 0.30 ^a^	10.36 ± 0.45 ^b^	0.035	0.544
n-3 fatty acid, g	1.85 ± 0.02	1.77 ± 0.0.04	1.61 ± 0.09	0.550	0.981
n-6 fatty acid, g	10.41 ± 0.11 ^a^	10.57 ± 0.27 ^a^	8.75 ± 0.37 ^b^	0.018	0.497
Cholesterol, mg	260.92 ± 2.94	256.80 ± 7.31	218.73 ± 12.69	0.337	0.947
Carbohydrate, g	309.02 ± 1.62	295.15 ± 3.59	284.30 ± 7.11	0.476	0.327
Dietary fiber, g	25.55 ± 0.19 ^a^	23.39 ± 0.35 ^b^	21.48 ± 0.59 ^c^	<0.001	<0.001
Calcium, mg	523.96 ± 4.30 ^a^	496.57 ± 8.73 ^ab^	449.91 ± 14.38 ^b^	0.027	0.058
Phosphate, mg	1118.68 ± 6.51	1056.07 ± 13.52	974.63 ± 30.62	0.057	0.015
Iron, mg	14.20 ± 0.18	13.42 ± 0.26	12.82 ± 0.50	0.675	0.774
Sodium, mg	3725.85 ± 31.75 ^a^	3450.53 ± 58.66 ^ab^	3174.20 ± 112.84 ^b^	0.026	0.012
Potassium, mg	3004.27 ± 19.72 ^a^	2809.08 ± 37.49 ^a^	2581.21 ± 68.39 ^b^	0.001	0.003
Carotene, μg	3200.00 ± 66.52	2916.37 ± 86.37	2881.59 ± 198.19	0.407	0.640
Retinol, μg	152.80 ± 6.41	137.92 ± 6.79	118.50 ± 13.09	0.183	0.245
Thiamin, mg	1.61 ± 0.01	1.52 ± 0.03	1.45 ± 0.05	0.363	0.288
Riboflavin, mg	1.60 ± 0.01 ^a^	1.52 ± 0.03 ^ab^	1.36 ± 0.04 ^b^	0.035	0.124
Niacin, mg	15.10 ± 0.11	14.31 ± 0.24	13.48 ± 0.52	0.498	0.656
Vitamin C, mg	76.16 ± 1.27	72.00 ± 2.41	65.55 ± 4.21	0.301	0.615

PHQ-9: Patient Health Questionnaire-9. PHQ depression severity was divided by PHQ total scores (normal: <5, mild: 5–9, moderate-to-severe: ≥10). Values are presented as mean ± standard error. ^a–c^ Significantly different at *p* < 0.05 by Bonferroni’s test. Adjusted for age, sex, body mass index, education level, alcohol drinking consumption, energy intake, disease history, and physical activity.

**Table 4 nutrients-15-01060-t004:** Scores of Korean healthy eating index of subjects according to diet quality level.

	Diet Quality Level				
	T1 (<58) (*n* = 4008)	T2 (58–69) (*n* = 4547)	T3 (≥70) (*n* = 4984)	*p*-Value	*p* Trend
Total score	48.20 ± 0.14 ^a^	63.46 ± 0.06 ^b^	76.91 ± 0.11 ^c^	<0.001	<0.001
Adequacy item					
Have breakfast	5.18 ± 0.08 ^a^	7.52 ± 0.06 ^b^	8.77 ± 0.05 ^c^	<0.001	<0.001
Mixed grain intake	1.22 ± 0.04 ^a^	2.02 ± 0.04 ^b^	2.92 ± 0.14 ^c^	<0.001	<0.001
Total fruits intake	1.06 ± 0.03 ^a^	2.11 ± 0.04 ^b^	3.40 ± 0.04 ^c^	<0.001	<0.001
Fresh fruits intake	1.16 ± 0.04 ^a^	2.33 ± 0.04 ^b^	3.66 ± 0.04 ^c^	<0.001	<0.001
Total vegetables intake	2.91 ± 3.56 ^a^	3.56 ± 0.02 ^b^	3.96 ± 0.02 ^c^	<0.001	<0.001
Vegetable intake excluding kimchi and pickled vegetables	2.52 ± 0.03 ^a^	3.28 ± 0.03 ^b^	3.82 ± 0.03 ^c^	<0.001	<0.001
Meats/fishes/eggs/beans intake	5.78 ± 0.07 ^a^	7.08 ± 0.05 ^b^	8.44 ± 0.04 ^c^	<0.001	<0.001
Milk and dairy product intake	1.80 ± 0.07 ^a^	2.88 ± 0.08 ^b^	5.23 ± 0.09 ^c^	<0.001	<0.001
Moderation item					
Energy from saturated fatty acid	5.78 ± 0.09 ^a^	7.97 ± 0.06 ^b^	8.74 ± 0.05 ^c^	<0.001	<0.001
Sodium intake	6.66 ± 0.07 ^a^	6.37 ± 0.06 ^b^	6.61 ± 0.06 ^a^	<0.001	<0.001
Energy from sweets/beverages	8.23 ± 0.06 ^a^	9.33 ± 0.03 ^b^	9.75 ± 0.02 ^c^	<0.001	<0.001
Balance item					
Energy from carbohydrate	1.52 ± 0.04 ^a^	2.49 ± 0.03 ^b^	3.41 ± 0.03 ^c^	<0.001	<0.001
Energy from fat	2.23 ± 0.04 ^a^	3.37 ± 0.03 ^b^	4.28 ± 0.03 ^c^	<0.001	<0.001
Total energy intake	2.13 ± 0.04 ^a^	3.15 ± 0.04 ^b^	3.93 ± 0.03 ^c^	<0.001	<0.001

Diet quality level was divided by KHEI total scores (T1: <58, T2: 58–69, T3: ≥70). Values are presented as mean ± standard error. ^a–c^ Significantly different at *p* < 0.05 by Bonferroni’s test. Adjusted for age, sex, body mass index, education level, alcohol drinking consumption, energy intake, disease history, and physical activity.

**Table 5 nutrients-15-01060-t005:** Nutrient intakes of subjects according to diet quality level.

	Diet Quality Level				
	T1 (<58) (*n* = 4008)	T2 (58–69) (*n* = 4547)	T3 (≥70) (*n* = 4984)	*p*-Value	*p* Trend
Protein, g	70.53 ± 0.55 ^a^	73.93 ± 0.48 ^b^	77.85 ± 0.39 ^c^	<0.001	<0.001
Fat, g	54.83 ± 0.59 ^a^	43.82 ± 0.38 ^b^	44.04 ± 0.29 ^b^	<0.001	<0.001
Saturated fatty acid, g	18.03 ± 0.24 ^a^	13.35 ± 0.15 ^b^	13.03 ± 0.11 ^b^	<0.001	<0.001
Monounsaturated fatty acid, g	18.28 ± 0.25 ^a^	13.76 ± 0.15 ^b^	13.77 ± 0.12 ^b^	<0.001	<0.001
Polyunsaturated fatty acid, g	12.39 ± 0.18 ^a^	11.82 ± 0.13 ^b^	12.31 ± 0.12 ^a^	0.002	0.932
n-3 fatty acid, g	1.67 ± 0.03 ^a^	1.78 ± 0.03 ^b^	2.04 ± 0.03 ^c^	<0.001	<0.001
n-6 fatty acid, g	10.73 ± 0.16 ^a^	10.04 ± 0.11 ^b^	10.28 ± 0.10 ^b^	0.002	0.044
Cholesterol, mg	247.74 ± 4.38 ^a^	250.50 ± 3.41 ^a^	274.94 ± 3.71 ^b^	<0.001	<0.001
Carbohydrate, g	277.23 ± 1.90 ^a^	315.29 ± 1.27 ^b^	326.04 ± 1.11 ^c^	<0.001	<0.001
Dietary fiber, g	20.60 ± 0.20 ^a^	25.06 ± 0.21 ^b^	29.61 ± 0.22 ^c^	<0.001	<0.001
Calcium, mg	441.32 ± 5.01 ^a^	509.74 ± 4.91 ^b^	598.01 ± 5.57 ^c^	<0.001	<0.001
Phosphate, mg	991.31 ± 6.12 ^a^	1093.09 ± 5.45 ^b^	1222.82 ± 5.38 ^c^	<0.001	<0.001
Iron, mg	12.23 ± 0.15 ^a^	14.52 ± 0.34 ^b^	15.33 ± 0.17 ^b^	<0.001	<0.001
Sodium, mg	3789.46 ± 40.45 ^a^	3745.11 ± 35.71 ^a^	3438.90 ± 29.66 ^b^	<0.001	<0.001
Potassium, mg	2554.59 ± 18.78 ^a^	2977.19 ± 21.22 ^b^	3341.89 ± 18.65 ^c^	<0.001	<0.001
Carotene, μg	2636.90 ± 95.42 ^a^	3188.43 ± 87.47 ^b^	3628.35 ± 71.26 ^c^	<0.001	<0.001
Retinol, μg	147.38 ± 11.86	139.72 ± 11.06	159.36 ± 5.50	0.120	0.497
Thiamin, mg	1.48 ± 0.02 ^a^	1.60 ± 0.01 ^b^	1.70 ± 0.01 ^c^	<0.001	<0.001
Riboflavin, mg	1.50 ± 0.01 ^a^	1.53 ± 0.01 ^a^	1.69 ± 0.01 ^b^	<0.001	<0.001
Niacin, mg	14.15 ± 0.14 ^a^	14.83 ± 0.12 ^b^	15.70 ± 0.11 ^c^	<0.001	<0.001
Vitamin C, mg	55.67 ± 1.13 ^a^	75.12 ± 1.88 ^b^	94.70 ± 1.96 ^c^	<0.001	<0.001

Diet quality level was divided by KHEI total scores (T1: <58, T2: 58–69, T3: ≥70). Values are presented as mean ± standard error. ^a–c^ Significantly different at *p* < 0.05 by Bonferroni’s test. Adjusted for age, sex, body mass index, education level, alcohol drinking consumption, energy intake, disease history, and physical activity.

**Table 6 nutrients-15-01060-t006:** Metabolic syndrome biomarkers of subjects according to depression severity.

	PHQ-9 Depression Severity				
	Normal (*n* = 10,888)	Mild (*n* = 1905)	Moderate to Severe (*n* = 746)	*p*-Value	*p* Trend
Waist circumference, cm	82.30 ± 0.13	80.60 ± 0.30	81.89 ± 0.53	0.188	0.086
Triglyceride, mg/dL	136.27 ± 1.42 ^a^	138.08 ± 3.46 ^b^	148.44 ± 5.81 ^b^	<0.001	<0.001
HDL-cholesterol, mg/dL	50.92 ± 0.17 ^a^	51.59 ± 0.33 ^b^	51.51 ± 0.58 ^ab^	0.015	0.020
Systolic blood pressure, mmHg	117.52 ± 0.22	115.12 ± 0.47	117.23 ± 0.70	0.085	0.061
Diastolic blood pressure, mmHg	76.13 ± 0.14 ^a^	74.55 ± 0.30 ^b^	74.33 ± 0.47 ^b^	0.008	0.003
Fasting blood glucose, mg/dL	99.52 ± 0.26	98.15 ± 0.57	102.24 ± 1.35	0.066	0.026

PHQ-9: Patient Health Questionnaire-9. PHQ depression severity was divided by PHQ total scores (normal: <5, mild: 5–9, moderate-to-severe: ≥10). Values are presented as mean ± standard error. ^a, b^ Significantly different at *p* < 0.05 by Bonferroni’s test. Adjusted for age, sex, body mass index, education level, alcohol drinking consumption, energy intake, and physical activity.

**Table 7 nutrients-15-01060-t007:** Adjusted odds ratios (95% confidence intervals) for risks of metabolic syndrome and its components of subjects according to depression severity.

	PHQ-9 Depression Severity			
	Normal (*n* = 10,888)	Mild (*n* = 1905)	Moderate to Severe (*n* = 746)	*p* Trend
Metabolic syndrome	1 (Reference)	1.01 (0.87–1.18)	1.47 (1.17–1.86) **	0.006
Abdominal obesity ^1^		1.13 (0.91–1.41)	1.15 (0.84–1.16)	0.201
Hypertriglyceridemia ^2^		1.14 (1.00–1.30) *	1.30 (1.05–1.60) *	0.003
Low HDL cholesterol ^3^		1.23 (1.08–1.39) **	1.22 (0.99–1.50)	0.001
High blood pressure ^4^		0.84 (0.73–0.97) *	0.89 (0.72–1.11)	0.034
Hyperglycemia ^5^		0.95 (0.84–1.08)	1.23 (1.00–1.50) *	0.205

PHQ-9: Patient Health Questionnaire-9. PHQ depression severity was divided by PHQ total scores (normal: <5, mild: 5–9, moderate-to-severe: ≥10). ^1^ waist circumference ≥ 90 cm for men, ≥80 cm for women. ^2^ Serum triglyceride level ≥ 150 mg/dL or current drug treatment for high triglyceride. ^3^ HDL-cholesterol < 40 mg/dL for men, 50 mg/dL for women. ^4^ Systolic blood pressure > 130 mmHg or diastolic blood pressure > 85 mmHg or current drug treatment for hypertension. ^5^ Fasting blood glucose ≥ 100 mg/dL or current drug treatment for a hypoglycemic agent or insulin. Values are presented as adjusted odds ratios (95% confidence intervals) adjusted for age, sex, body mass index, education level, drinking consumption, energy intake, and physical activity. * *p* < 0.05, ** *p* < 0.01.

**Table 8 nutrients-15-01060-t008:** Adjusted odds ratios (95% confidence intervals) for risks of metabolic syndrome and its components of subjects according to depression severity after stratified by diet quality levels measured by the Korean Healthy Eating Index.

	Diet Quality Levels											
	T1 (<58) (*n* = 4008)				T2 (58–69) (*n* = 4547)				T3 (≥70) (*n* = 4984)			
	PHQ-9 Depression Severity				PHQ-9 Depression Severity				PHQ-9 Depression Severity			
	Normal (*n* = 3035)	Mild (*n* = 636)	Moderate to severe (*n* = 337)	*p* trend	Normal (*n* = 3655)	Mild (*n* = 670)	Moderate to severe (*n* = 222)	*p* trend	Normal (*n* = 4198)	Mild (*n* = 599)	Moderate to Severe (*n* = 187)	*p* trend
Metabolic syndrome	1 (Reference)	0.98 (0.75–1.29)	1.72 (1.24–2.40) **	0.009	1 (Reference)	1.01 (0.79–1.31)	1.18 (0.78–1.77)	0.510	1 (Reference)	1.08 (0.82–1.42)	1.42 (0.90–2.25)	0.145
Abdominal obesity ^1^		1.16 (0.77–1.74)	1.699 (1.04–2.81) *	0.040		1.15 (0.80–1.64)	0.78 (0.45–1.38)	0.857		1.00 (0.68–1.47)	0.93 (0.55–1.56)	0.827
Hypertriglyceridemia ^2^		1.04 (0.83–1.31)	1.31 (0.96–1.78)	0.123		1.16 (0.96–1.41)	1.05 (0.74–1.48)	0.310		1.28 (1.00–1.64)	1.50 (1.00–2.26)	0.010
Low HDL cholesterol ^3^		1.35 (1.06–1.71) *	1.51 (1.11–2.03) **	<0.001		1.24 (1.01–1.52) *	1.03 (0.71–1.49)	0.219		1.10 (0.89–1.36)	1.10 (0.76–1.61)	0.361
High blood pressure ^4^		0.80 (0.65–1.06)	0.94 (0.67–1.30)	0.328		0.89 (0.70–1.12)	0.57 (0.41–0.80) *	0.003		0.84 (0.66–1.07)	1.35 (0.87–2.11)	0.889
Hyperglycemia ^5^		0.99 (0.79–1.25)	1.05 (0.78–1.42)	0.807		0.82 (0.66–1.02)	1.46 (1.02–2.10) *	0.513		0.80 (0.63–1.02)	1.27 (0.81–1.99)	0.264

PHQ-9: Patient Health Questionnaire-9. PHQ-9 depression severity was divided by total scores of PHQ-9 (normal: <5, mild: 5–9, moderate-to-severe: ≥10). KHEI: Korean Healthy Eating Index. Diet quality level was divided into tertiles by total scores of KHEI (Low (T1): <58, Medium (T2): 58–69, High (T3): ≥70). ^1^ Waist circumference ≥ 90 cm for men, ≥ 80 cm for women. ^2^ Serum triglyceride level ≥ 150 mg/dL or current drug treatment for high triglyceride. ^3^ HDL-cholesterol < 40 mg/dL for men, 50 mg/dL for women. ^4^ Systolic blood pressure > 130 mmHg or diastolic blood pressure > 85 mmHg or current drug treatment for hypertension. ^5^ Fasting blood glucose ≥ 100 mg/dL or current drug treatment for a hypoglycemic agent or insulin. Values are presented as adjusted odds ratios (95% confidence intervals) adjusted for age, sex, body mass index, education level, drinking consumption, energy intake, and physical activity. * *p* < 0.05, ** *p* < 0.01.

**Table 9 nutrients-15-01060-t009:** The mediating effect of diet quality on the association between depression severity and metabolic syndrome.

	Metabolic Syndrome (Dependent Variable, Y)		
	Model 1 X -> Y	Model 2 X + M -> Y	Proportion of Mediation
Depression severity (Independent Variable, X)	1.47 (1.17–1.86) **	1.45 (1.15–1.84) **	Fail to meet the conditions
Diet quality (Mediator, M)		0.89 (0.77–1.02)	

Values are presented as adjusted odds ratios (95% confidence intervals) adjusted for age, sex, body mass index, education level, drinking consumption, energy intake, and physical activity. ** *p* < 0.01.

## Data Availability

The data are available from the Korean National Health and Nutrition Examination Survey (KNHANES) website at https://knhanes.kdca.go.kr/knhanes/main.do (accessed on 1 March 2021).

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
