# Peer review of "Does Better Diet Quality Offset the Association between Depression and Metabolic Syndrome?"

_nutrients, 2023, doi:10.3390/nu15041060_

Round 1

Reviewer 1 Report

This is a nicely conducted analysis to examine whether a healthy diet quality could offset the positive relationship between depression and MetS in the general Korean adult population. Before considering for acceptance, I have some suggestions for authors on the analysis plan.

First, authors postulated that the cooccurrence of depression and MetS may be mediated by unhealthy lifestyle factors, including poor diet, and physical inactivity such as sedentary lifestyle. However, the analysis plan does not cover the mediation analysis. It is unclear the extent of depression-MetS association as explained by the unhealthy diet. A mediation analysis or pathway analysis is therefore necessary.

Second, it is unclear how different the dietary quality can be according to tertiles. T3 is greater than T1 by at least 12 points, so what does that mean? At least in a way as supplementary information, authors should compare the diet (according to the dietary recommendation of dietary quality index) among these three groups of participants, so readers would understand how participants are being classified. For example, what’s the recommendation for the item “Vegetable intake excluding Kimchi and pickled vegetables”? More explanation is needed.

Lastly, the subgroup, mediation and/or pathway analysis using dietary quality index should include the 14 subcomponents too, e.g. according to fulfilling ‘mixed grains intake’ or not, how does the relationship between depression and MetS differ? How does mixed grains intake may mediate the relationship? That would provide more solid evidence to demonstrate how a healthy dietary pattern would work to offset the depression-MetS relationship.

Author Response

Point 1: First, authors postulated that the cooccurrence of depression and MetS may be mediated by unhealthy lifestyle factors, including poor diet, and physical inactivity such as sedentary lifestyle. However, the analysis plan does not cover the mediation analysis. It is unclear the extent of depression-MetS association as explained by the unhealthy diet. A mediation analysis or pathway analysis is therefore necessary.

Response 1: Thank you for your valuable comments. Korean Healthy Eating Index (KHEI) was developed to evaluate the overall diet quality of Korean adults using a total score. However, based on your comments, we analyzed the mediating effects of diet quality (page 4, methods) as shown in Figure 2 and Table 9. Based on our results, no mediation effect existed including sub-categories of KHEI (Supplementary Table 1). Furthermore, a potential mediation effect of fulfilling each component of KHEI was examined (Supplementary Table 2) and mediating effects existed for vegetable subcomponents (proportion of mediation: total vegetable in-take 63.3%; vegetable intake excluding kimchi and pickled vegetables 66.3%). Therefore, scores of KHEI and nutrient intakes were compared according to fulfilling both vegetable intake subcomponents (Supplementary Table 3-4). Interestingly, having full scores of vegetable intakes had less scores of intakes of sodium and milk and dairy product and more sodium intakes than their counterpart. The positive associations between depression severity and risk of MetS was only observed in those without fulfilling vegetable intake subcomponents (Supplementary Table 5). However, an inverse association between depression severity and risk of high blood pressure existed in those with full scores of vegetable intake subcomponents (p trend = 0.047).

Point 2: Second, it is unclear how different the dietary quality can be according to tertiles. T3 is greater than T1 by at least 12 points, so what does that mean? At least in a way as supplementary information, authors should compare the diet (according to the dietary recommendation of dietary quality index) among these three groups of participants, so readers would understand how participants are being classified. For example, what’s the recommendation for the item “Vegetable intake excluding Kimchi and pickled vegetables”? More explanation is needed.

Response 2: We added the table comparing the components of Korean Healthy Index (KHEI) according to diet quality groups (T1, T2, and T3), as shown in Tables 4 and 5. And we also explained how the difference of score can be interpretated between groups based on the our KHEI development paper (doi: 10.4162/nrp.2022.16.2.233) as shown in page 7.

“As described [29], the difference in scores needs to be interpreted as a KHEI scoring system according to a dietary guideline for Koreans. For example, a 1.7 point difference between T3 and T1 in mixed grain intake means that T3 consumes 0.1 serving/day more mixed grains than T1 (maximum score: ≥ 0.3 serving/day).”

Point 3: Lastly, the subgroup, mediation and/or pathway analysis using dietary quality index should include the 14 subcomponents too, e.g. according to fulfilling ‘mixed grains intake’ or not, how does the relationship between depression and MetS differ? How does mixed grains intake may mediate the relationship? That would provide more solid evidence to demonstrate how a healthy dietary pattern would work to offset the depression-MetS relationship.

Response 3: Thank you for your valuable comments. As we mentioned in the response 1, Korean Healthy Eating Index (KHEI) was developed to evaluate the overall diet quality of Korean adults using a total score. However, based on your comments, we analyzed the mediating effects of diet quality (page 4, methods) as shown in Figure 2 and Table 9. Based on our results, no mediation effect existed including sub-categories of KHEI (Supplementary Table 1). Furthermore, a potential mediation effect of fulfilling each component of KHEI was examined (Supplementary Table 2) and mediating effects existed for vegetable subcomponents (proportion of mediation: total vegetable in-take 63.3%; vegetable intake excluding kimchi and pickled vegetables 66.3%). Therefore, scores of KHEI and nutrient intakes were compared according to fulfilling both vegetable intake subcomponents (Supplementary Table 3-4). Interestingly, having full scores of vegetable intakes had less scores of intakes of sodium and milk and dairy product and more sodium intakes than their counterpart. The positive associations between depression severity and risk of MetS was only observed in those without fulfilling vegetable intake subcomponents (Supplementary Table 5). However, an inverse association between depression severity and risk of high blood pressure existed in those with full scores of vegetable intake subcomponents (p trend = 0.047).

Reviewer 2 Report

I read the paper “Does better diet quality offset association between depression and metabolic syndrome?by In Seon Kim et al.

The manuscript is quite easy to ready. Statistical analysis is well performed.

Comments:

1.      Patients with mental health disorders do presents important eating disorders. Moreover, the quality of the food becomes poor. When anxiety and depression prevail in your life, everything which takes a minimum strength is postponed. It is possible that most of the patients eats industrial food and do not make food for themselves. Which might as well impact on the metabolite absorption and of salt intake (generally used to keep food and possibly in line with what you report). Do you have such data?

2.      The mildly depressed group presented with increase in triglycerides levels, which could be justified by an increase in insulin resistance, a crucial point in metabolic syndrome development, as well as diabetes and NAFLD (doi: 10.37349/emed.2020.00019). Please discuss this point. In fact, the food intake could lead to an insulin resistance worsening.

3.      The lack of data about drug therapy is a major limitation. In fact, several drugs used for depression treatment may lead to an increase in body weight, thus worsening metabolic syndrome. As well as some drugs, which play a role in reducing insulin resistance, such as metformin, have not been investigated (doi: 10.1001/jama.2019.11489). Please report in the appropriate section and discuss it.

Author Response

Point 1: Patients with mental health disorders do presents important eating disorders. Moreover, the quality of the food becomes poor. When anxiety and depression prevail in your life, everything which takes a minimum strength is postponed. It is possible that most of the patients eats industrial food and do not make food for themselves. Which might as well impact on the metabolite absorption and of salt intake (generally used to keep food and possibly in line with what you report). Do you have such data?

Response 1: Thank you for your valuable comments. We added the table that compared dietary intakes (including sodium) according to depression severity in Table 3. Moreover, we explained the association between depression, MetS and sodium in results and discussion.

“Whether diet quality mediated the association between depression se-verity and risk of MetS was also tested controlling for the potential con-founders (Table 9, Figure 2). However, no mediation effect existed including sub-categories of KHEI (Supplementary Table 1). Furthermore, a potential mediation effect of fulfilling each component of KHEI was examined (Supplementary Table 2) and mediating effects existed for vegetable sub-components (proportion of mediation: total vegetable intake 63.3%; vege-table intake excluding kimchi and pickled vegetables 66.3%). Therefore, scores of KHEI and nutrient intakes were compared according to fulfilling both vegetable intake subcomponents (Supplementary Table 3-4). Interestingly, having full scores of vegetable intakes had less scores of intakes of sodium and milk and dairy product and more sodium intakes than their counterpart. The positive associations between depression severity and risk of MetS was only observed in those without fulfilling vegetable intake sub-components (Supplementary Table 5). However, an inverse association between depression severity and risk of high blood pressure existed in those with full scores of vegetable intake subcomponents (p trend = 0.047).”

“Most of the scores of KHEI subcomponents were lower in the two de-pression groups than the normal group, but it is worth noting that the score of sodium intake was greater due to a lower intake of sodium in the two groups. Furthermore, an inverse association between depression severity and risk of high blood pressure existed in the middle tertile group of diet quality and in those with full scores of vegetable intake subcomponents. The same inverse association has been found in the cross-sectional studies in the general US female population using National Health and Nutrition Examination Survey [33] and elderly Japanese women [34]. Excessive sodium intake increases risks of hypertension and cardiovascular events due to negative physiological effects on body system [35]. Due to the nature of cross-sectional studies including ours, it is not clear whether reduction in sodium intake increases stress or stress increases sodium intake. However, considering mediating effects of fulfilling vegetable subcomponents of KHEI, stress may increase sodium intake via high vegetable intake that eventually increases risk of MetS. Further understanding between depression, diet quality, and MetS is needed.”

Point 2: The mildly depressed group presented with increase in triglycerides levels, which could be justified by an increase in insulin resistance, a crucial point in metabolic syndrome development, as well as diabetes and NAFLD (doi: 10.37349/emed.2020.00019). Please discuss this point. In fact, the food intake could lead to an insulin resistance worsening.

Response 2: We added the information of relationship between triglyceride level and insulin resistance in discussion.

“Among components of MetS, no association between depressive symptoms and hyperglycemia existed. Although hyperglycemia plays a crucial role in MetS development as an indicator of insulin resistance, it is known that an elevated TG levels is also correlated with insulin resistance in individuals with normal glucose levels [41]. Insulin resistance may cause nonalcoholic fatty liver disease by elevated accumulation of free fatty acids due to hypertriglyceridemia in the liver [42, 43] before type 2 diabetes de-velops [44, 45]. Therefore, individuals with dyslipidemia needs to be treat-ed carefully, even though their blood glucose levels are still normal.”

Point 3: The lack of data about drug therapy is a major limitation. In fact, several drugs used for depression treatment may lead to an increase in body weight, thus worsening metabolic syndrome. As well as some drugs, which play a role in reducing insulin resistance, such as metformin, have not been investigated (doi: 10.1001/jama.2019.11489). Please report in the appropriate section and discuss it.

Response 3: Thank you for your comments. Based on your comments, we additionally discussed about the drugs, which could be affect the status of depression and metabolic syndrome in discussion.

                 “Lastly, the lack of data about medication use may be also a major limitation. Metformin, a major medication for type 2 diabetes, has been proposed to treat comorbid depression and diabetic patients [68]. Several studies have suggested that antidepressants can induce (Mirtazapine (SNRI), TCAs am-itriptyline and nortriptyline) and reduce (selective Serotonin re-uptake inhibitors) weight gain as a side effect [69-71], although results vary de-pending on the duration of exposure and types of medication [72]. There-fore, information of medication needs to be included in future studies.”

Round 2

Reviewer 1 Report

Authors have addressed my comments adequately, I have no further comments.

Reviewer 2 Report

The authors addressed all the issue I raised. I am satisfied.